# Functioning of the EROS-R Scale in a Clinical Sample of Psychiatric Patients: New Psychometric Evidence from the Classical Test Theory and the Item Response Theory

**DOI:** 10.3390/ijerph191610062

**Published:** 2022-08-15

**Authors:** Lindsey W. Vilca, Evelyn L. Chambi-Mamani, Emely D. Quispe-Kana, Mónica Hernández-López, Tomás Caycho-Rodríguez

**Affiliations:** 1South American Center for Education and Research in Public Health, Universidad Norbert Wiener, Lima 15011, Peru; 2Department of Psychology, Universidad Peruana Unión, Lima 15472, Peru; 3Department of Psychology, Universidad de Jaén, 23071 Jaén, Spain; 4Facultad de Ciencias de la Salud, Carrera de Psicología, Universidad Privada del Norte, Lima 15314, Peru

**Keywords:** behavioral activation, Environmental Reward Observation Scale, depression, anxiety, clinical sample

## Abstract

Reliable and valid assessment instruments that can be applied briefly and easily in clinical and outpatient settings that provide information about the sources of reinforcement that the patient finds in his life are especially relevant in therapy. The study aimed to evaluate the psychometric properties of the Environmental Reward Observation Scale (EROS-R) in a sample of psychiatric patients. A sample of 228 psychiatric patients of both sexes (56.1% men and 43.9% women) aged between 18 and 70 years was selected. Along with the EROS-R, other instruments were administered to assess depression and anxiety. The results show that the scale fits a unidimensional model, presenting adequate fit indices (RMSEA = 0.077 (IC 90% 0.055–0.100); SRMR = 0.048; CFI = 0.98; TLI = 0.98). It was also shown that the degree of reward provided by the environment (EROS-R) correlates negatively with the level of depression (*ρ* = −0.54; *p* < 0.01) and anxiety (*ρ* = −0.34; *p* < 0.01). From the IRT perspective, all the items present adequate discrimination indices, where item 4 is the most precise indicator to measure the degree of environmental reward. All this leads us to conclude that the EROS-R is an instrument with robust psychometric guarantees from TCT and IRT’s perspectives, making it suitable for use in clinical contexts.

## 1. Introduction

Behavioral activation (BA) is a therapeutic approach encompassed in so-called third-generation therapies [1]. Its efficacy has been demonstrated in several systematic reviews and meta-analysis studies in children and adolescents [2], adults [3,4], older adults [5] and in people with neurological conditions [6]. At a theoretical level, it starts from a basic premise: the key to changing how a person feels begins by helping them change what they do [7]. This idea is perfectly consistent with classical behavioral models of depression that assume that when people have a low level of positive reinforcement or a high level of punishment, they can develop depressive behavior patterns [8]. If people behave and do not obtain desirable results, they are likely to avoid punitive or non-reinforcing contexts and stop doing those things that are important to them. The less they do, the less they want to do (or feel they cannot do), entering a vicious circle. The withdrawal from life leads them to focus on how they feel. Their struggle now centers on avoiding bothersome thoughts and feelings and the conditions that cause them. By failing to act and follow what matters to them in life, they reduce contact with potential reinforcers in the environment, thus increasing their feelings of sadness and making problems worse in a characteristic pattern of experiential avoidance [9]. BA proposes, through a brief and structured program: (1) to increase behaviors that allow the person to be in an environment that provides reinforcing contingencies and (2) to break with the dysfunctional pattern of experiential avoidance [10,11]. It is precisely its brief and structured nature and the relative simplicity of its administration that make BA a cost-effective intervention approach whose adoption for the treatment of psychiatric inpatients with depressive symptomatology has produced positive results [12,13,14]. These positive results appear to be preceded by an increase in reinforcement contingencies provided by the environment [15,16], which is coherent with the theory underlying BA. It is well-known that an increase in the sources of reinforcement subsequently produces an increase in psychiatric inpatients’ motivation, facilitating the development of new skills and the pursuit of personally relevant goals, as well as an increase in their levels of self-efficacy [17].

One of the keys to treatment in BA is the use of records in which patients monitor the activities they carry out, as well as the variations that getting involved in these activities generates in their state of mind and in the emotions they experience [18]. These records serve as an evaluation tool and function as an active part of the therapy [7] and, therefore, are irreplaceable. However, reliable and valid evaluation instruments that can be applied briefly and easily in clinical and outpatient settings and that provide information about the sources of reinforcement the patient finds in his life can be a particularly relevant complement to assessing the changes resulting from activation.

In this sense, the Environmental Reward Observation Scale [19] (EROS) is one of the most used instruments to measure the behavioral change in patients and monitor the objectives set in BA. The EROS is a brief ten-item scale that, according to its authors, makes it possible to measure the availability of reinforcement and the person’s ability to obtain positive reinforcement from their environment. The scale has been widely used both in correlational studies that explore the relationship between reinforcement availability and symptomatology [20,21,22,23] and in intervention studies focused on determining the effectiveness of treatments and their processes of change [16,24,25,26,27,28]. However, it is essential to note that the psychometric properties of the EROS were originally explored with a non-clinical population, and there is hardly any evidence of the psychometric properties of the scale and its operation in a clinical population (patients with a diagnosis of depression, anxiety, psychosis, or other psychiatric disorders).

Barraca and Pérez-Álvarez [29] carried out the first adaptation of the instrument for a Spanish population and found that the scale had adequate internal consistency. Conducting an exploratory factor analysis, they found a two-factor solution; although the first factor explained more than 45% of the variance, they concluded that the unifactorial structure of the original instrument could be accepted. In addition, they found that the EROS presented significant correlations with measures of anxiety, depression, and experiential avoidance and made it possible to discriminate between clinical and non-clinical samples, but not between clinical samples with different diagnoses. Recently, Fernández-Rodríguez et al. [30] validated the instrument with a sample of female breast cancer survivors. However, from the classical test theory (CTT) perspective, the study showed that the one-dimensional model presented adjustment problems (CFI = 0.95; RMSEA = 0.116 (IC 90% 0.096–0.137)) and from the perspective of the item response theory (IRT), items 2 (*a* = 0.72) and 10 (*a* = 0.95) presented low levels of discrimination. Additionally, with a Spanish-speaking population, Vilca et al. [31] developed a version of only positive items (EROS-R) of the EROS scale. This instrument presented more robust psychometric properties compared to the original positive and negative item scale. In addition, it showed excellent psychometric properties in internal structure-based validity (RMSEA = 0.07; CFI = 0.98; TLI = 0.98), relationship-based validity with other variables, and test–retest reliability (0.70 (CI 95% 0.593–0.788)). It also showed strict factorial invariance according to the sex of the participants (Δχ^2^ = 0.34; *p* > 0.05; ΔCFI = 0.003) and over time (Δχ^2^ = 0.29; *p* > 0.05; ΔCFI = 0.004). However, the psychometric properties of the EROS-R scale were studied in a non-clinical population, specifically, university students. Therefore, psychometric evidence of the EROS-R scale has not yet been reported using item response theory (IRT) in a clinical population.

It is essential to point out that IRT models have three fundamental advantages that can be used in the clinical context [32]: (a) Invariance of the item parameters; that is, the item parameters do not vary, although the people who answer are different. (b) Invariance of the evaluated trait parameter concerning the instrument used for its estimation; that is, the person’s skill level does not depend on the test. (c) It provides local precision measures through the item information curve (IIC) and the test information curve (TIC). These characteristics allow knowing in detail the area in which the trait measured by the test is being measured better. In other words, it allows knowing for which trait level the instrument is best designed. These characteristics also allow for more reliable comparisons between those evaluated.

Based on the preceding, evaluating and guaranteeing the proper psychometric performance of the scale in clinical samples is essential. We believe the scale will be helpful in applied contexts and with patients with psychological disorders, mainly for two reasons. First, the scale provides information that could be used to make clinical decisions about the process and the extent of treatment. Second, it could allow a better understanding of how patients, in a clinical context, perceive the availability of positive reinforcement from their environment and how this variable could mediate intervention results.

For all the above, the present study was carried out with the following objectives: (a) to evaluate the validity based on the internal structure through CFA and IRT models, (b) to evaluate the validity based on its relationship with other variables, and (c) to estimate the reliability of the scale in psychiatric patients.

## 2. Method

### 2.1. Design

The present study has an instrumental design since the objectives are oriented to the evaluation of the psychometric properties of a measurement instrument [33].

### 2.2. Participants

A sample of 228 Peruvian psychiatric patients of both genders (128 (56.1%) male and 100 (43.9%) female) between the ages of 18 and 70 years (*M* = 42.1, *SD =* 14.1) who attended two mental health centers as outpatients were collected. The most frequent diagnoses were schizophrenia (152 (66.5%)), bipolar disorder (35 (15.4%)), major depression (14 (6.1%)), borderline personality disorder (11 (4.6%)), and obsessive compulsive disorder (7 (3.1%)). Regarding their educational level, the majority completed secondary school (122 (53.5%)) and the others, to a lesser extent, completed primary school (64 (28.1%)) and higher education (42 (18.4%)). Regarding the treatment time received in both centers, the highest proportion of patients received one month to one year of treatment (74.1%). Furthermore, all participants lived in an urban area.

The following exclusion criteria were used: (a) active suicidal ideation, (b) moderate or severe neurological deficit, (c) active symptoms of psychosis that do not allow adequate contact with reality, and (d) medical conditions that prevent the patient from participating in the study.

## 3. Measures

### 3.1. Environmental Reward Observation Scale (EROS-R)

The original EROS scale was developed by Armento and Hopko [19] to measure the degree of reward provided by the environment. For the present study, the version adapted into Spanish by Vilca et al. [31], where it showed adequate reliability indices (ω = 0.93 (IC 95% = 0.92–0.94)) and validity based on the internal structure (CFI = 0.98; TLI = 0.98; RMSEA = 0.075) in university students, was used. The scale is made up of 10 items (Appendix A) that are answered with a Likert-type scale with four response categories ranging from: “totally disagree” (1) to “totally agree” (4), and where a higher score shows a greater experience of reinforcing experiences from the environment. The version used does not have reverse items.

### 3.2. Generalized Anxiety Disorder Scale (GAD-7)

The GAD-7 instrument was developed by Spitzer et al. [34] to measure generalized anxiety in primary care settings. The version adapted into Spanish by García-Campayo et al. [35] was used for the study, where the one-dimensional model presented adequate fit indices (AGFI = 0.91; GFI = 0.96; CFI = 0.99; NFI = 0.98; RMSEA = 0.080) and internal consistency (α = 0.94). The scale consists of seven items that are answered using a Likert-type scale with four response categories ranging from “never” (0) to “almost every day” (3), where a higher score indicates a greater presence of the disorder. In the present study, the one-dimensional model presented adequate reliability indices (α = 0.88; ω = 0.85).

### 3.3. Patient Health Questionnaire (PHQ-9)

Developed by Spitzer et al. [36] and adapted to Spanish by Zhong et al. [37], the PHQ-9 aims to detect depression in primary care settings. The version adapted into Spanish demonstrated unidimensionality, local independence, and an acceptable fit for the Rasch IRT model [37]. The questionnaire is made up of nine items that have four categories ranging from “not at all” (0) to “almost every day” (3). A higher score on the questionnaire indicates the presence of depression. In the present study, the one-dimensional model presented adequate reliability indices (α = 0.84; ω = 0.81).

### 3.4. Procedure

The study obtained the approval of the ethics committee of the Center for Research and Innovation in Health (CIISA) of the Universidad Peruana Unión (2021-CE-FCS-UPeU-00227). In addition, the study complied with the standards established by the Declaration of Helsinki [38].

For data collection, non-probabilistic sampling was used for convenience. Two clinical psychologists, who previously coordinated with the two mental health centers, developed the list of possible participants. For the application of the instruments, a specific session was scheduled, directed by the two clinical psychologists, and lasted approximately 20 min. It is important to point out that the evaluation of the patients was carried out individually. Before application, the legal guardians were informed of the study’s objectives, and their permission to participate in the study was obtained. During the data collection, the anonymity and confidentiality of the results were ensured. In addition, the study’s objectives were explained to the patients, and doubts related to the procedure were resolved. A previous evaluation was also carried out to determine the attention and understanding capacity of the patients. Only lucid patients who had their legal guardians’ approval and consent to participate in the study completed the instruments. It is important to note that the instruments were self-administered by the patients themselves.

### 3.5. Data Analysis

A confirmatory factor analysis (CFA) was performed using the diagonally weighted least squares with mean- and variance-corrected (WLSMV) estimator since the items are at the ordinal level [39]. The root mean square error of approximation (RMSEA), standardized root mean squared residual (SRMR), comparative fit index (CFI), and Tucker–Lewis index (TLI) were used to evaluate the model’s fit. For the RMSEA and SRMR indices, values less than 0.08 were considered acceptable [40]. For the CFI and TLI, values greater than 0.95 were considered adequate [41]. To evaluate the internal consistency of the scale, Cronbach’s alpha coefficient [42], the omega coefficient corrected for correlated errors [43] and the omega coefficient for ordinal items [44] were used, where a value of ω > 0.80 was considered appropriate [45].

For item response theory (IRT), a graded response model [46] was used, specifically, an extension of the 2-parameter logistic model (2-PLM) for ordered polytomous items [47]. The M2* test was used to estimate the fit of the one-dimensional model [48], and the following fit criteria were used: RMSEA ≤ 0.08 and SRMSR ≤ 0.05 [49]. The CFI and TLI values were also taken into account, using the same adjustment criteria (≥0.95) used in SEM models [50]. Additionally, a generalization of the S-X2 estimator for polytomous items was used to assess the fit of the items [51]. Items with a *p*-value < 0.001 showed poor fit [52]. In addition, the size of the RMSEA was evaluated, where small values (<0.06) showed a good fit [53].

Two parameters were estimated for each item: discrimination (a) and location (b). The discrimination parameter (a) determines the slope by which responses to items change as a function of the level in the latent trait, and the item location parameters (b) determine how much of the latent trait the item requires to be answered. Since the scales have four response categories, there are three location estimates per threshold. Estimates for these three thresholds indicate the latent variable level at which an individual has a 50% chance of scoring at or above a particular response category. The item and test information curves (IIC and TIC, respectively) were also calculated.

An SEM model was used for the validity based on the relationship with other variables. In this model, the degree of reward provided by the environment is related to the level of anxiety and depression. The WLSMV estimator was used to estimate the model, and the same adjustment indicators made in the confirmatory factor analysis (CFA) were taken into account.

All statistical analyses were performed using the “lavaan” package [54] and the “ltm” package for the GRM [55]. Both cases used the RStudio environment [56] in R [57].

## 4. Results

### 4.1. Descriptive Analysis

It can be seen in Table 1 that in the polychoric correlation matrix, all the items have a moderate correlation coefficient. It can also be seen that item 10 (“The activities I do normally go well”) presents the highest average score (*M* = 2.70), whereas item 5 (“My life is as rewarding as other people’s”) presents the lowest average score (*M* = 2.52) in the clinical sample. Regarding the asymmetry and kurtosis indices, it can be seen in Table 1 that the items present adequate indices.

### 4.2. Validity Based on Internal Structure

The results of the study show evidence in favor of a one-dimensional model (χ^2^ = 75.60; df = 32; *p* = 0.000; RMSEA = 0.077 (IC 90% 0.055–0.100); SRMR = 0.048; CFI = 0.98; TLI = 0.98). Figure 1 shows that the factorial weight of the latent variable with each of its observed variables is high and significant. In addition, following the method Saris et al. [58] used to evaluate the relevance of the modification indices (MI) and according to the content analysis of items, a correlation was specified between the errors of items 1 and 2 (0.64), 8 and 9 (0.27), and 9 and 10 (0.45).

### 4.3. Reliability

The scale shows adequate levels of reliability since it presents an adequate corrected omega coefficient (ω_c_ = 0.87). Similarly, it presents an adequate ordinal omega (ω_o_ = 0.88) and a Cronbach’s alpha coefficient that can be considered acceptable (α = 0.89).

### 4.4. Item Response Theory Model

The results found in the confirmatory factor analysis (CFA) allow us to fulfill the unidimensionality assumption. However, the presence of correlated errors in three pairs of items in the CFA model indicates a possible violation of the local independence assumption. To assess the impact of this violation on the item parameters, we first examined the discrimination values of the three pairs of items in the IRT model. As can be seen in Table 2, the discrimination parameters for these five items were not excessively high (1.48 to 2.05) and, on the contrary, they presented the lowest values in the range of model scores (1.48 to 3.35). Second, a sensitivity analysis was performed to examine the change in discrimination values by removing items 2 and 9. The range of item values was very similar to the original model results (1.48 to 3.35 for the 10 items; 1.43 to 3.37 without items 2 and 9). In addition, the order of classification of the items according to their discrimination values was almost identical, and items 1 (1.46), 8 (1.43), and 10 (1.89) continued to present the lowest discrimination values. Finally, the changes in the discrimination parameters between the two models were minimal (<0.16). Taking all of the above into account, it can be stated that the original ten-item model is sufficiently robust against correlated errors in the CFA model.

A gradual response model (GRM) was used to estimate the model, specifically, an extension of the 2-parameter logistic model (2-PLM) for ordered polytomous items. Table 2 shows that the GRM model presents adequate fit indices (M2 (df) = 29.05 (15); *p* = 0.016; RMSEA = 0.064; SRMSR = 0.068; TLI = 0.95; CFI = 0.97). In addition, it is observed in Table 2 that all the items present *p*-values associated with an S-χ^2^ less than 0.001 and small RMSEA values (<0.06). Therefore, it can be affirmed that the items present adequate fit indices in the GRM model. Concerning the parameters of the model, in Table 2, it can be seen that all the discrimination parameters of the items are above the value of 1, which is generally considered good discrimination [47]. Regarding the location parameters, all the threshold estimators increased monotonically. A greater presence of the latent trait is required to answer the higher response categories.

Figure 2 shows the item and test information curves (IIC and TIC, respectively). In the IIC, it can be seen that item 4 is the most precise of the scale to evaluate the latent trait. In addition, the TIC shows that the test is more reliable (accurate) in the scale range between −2.5 and 2.5.

### 4.5. Validity Based on the Relationship to Other Constructs

Considering the literature review, an SEM model was proposed to evaluate the latent relationship between the EROS scale and the level of anxiety and depression. It can be seen in Table 3 that the structural model presents adequate adjustment indices (χ^2^ = 581.03; *df* = 293; *p* = 0.000; RMSEA = 0.066 (IC 90% 0.058–0.074); SRMR = 0.086; CFI = 0.92; TLI = 0.91) and the measurement models are adequately represented by their items.

Figure 3 shows that the degree of reward provided by the environment (EROS-R) is negatively related to the level of depression (*ρ* = −0.54; *p* < 0.01) and anxiety (*ρ* = −0.34; *p* < 0.01). Taking these results into account, it can be concluded that the scale has validity based on the relationship with other constructs.

## 5. Discussion

The main objective of the study was to evaluate the psychometric properties of the EROS-R scale in psychiatric patients; in this sense, EROS is one of the most used instruments to evaluate the effectiveness of BA [23,25,28,59,60,61,62]; however, data on its psychometric performance in the clinical context are limited. The adaptation and validation of the scale to Spanish revealed some psychometric problems [29,30]. In contrast, the recent validation of the EROS-R instrument in Spanish, with only positive items, showed better psychometric performance [31]. However, this version has not been validated with a clinical population. The present study is the first to evaluate the psychometric properties of the EROS-R scale in psychiatric patients. The results show that this version of the EROS, with only positive items, has robust psychometric guarantees from the perspective of TCT and IRT, making it suitable for clinical use.

Regarding the factorial structure of the scale, the results of the CFA confirmed the presence of a one-dimensional model, thus replicating the model of the original scale [19] and the validation results of the EROS-R, both developed with the general population. Additionally, some correlated errors between the items could be attributed to similar conceptual content [39]. These correlated errors (1~2, 8~9, 9~10) are similar to those reported in the study by Vilca et al. [31], except for the correlation of items 9 (“My life is interesting”) and 10 (“The activities I do normally go well”). Both items share a positive evaluative relationship, referring to positive affectivity in one’s personal experience. Although these items have the lowest factorial weights in the model, they are still adequate since their values are above 0.60 [63,64,65]. In other words, the factor manages to extract enough variance from the items; therefore, these items manage to represent the construct adequately. Future studies should explore these results with different samples. In any case, it is essential to point out that these results constitute the first psychometric evidence of the existence of a unidimensional model of the EROS-R scale in the clinical population, both in Spanish-speaking countries and in the rest of the world.

Concerning the scale’s reliability, although the α coefficient shows an adequate level of internal consistency, different researchers have pointed out that this coefficient can produce biased estimates in the presence of correlated errors, as is the case in Gu et al. [66]. In addition, its use is not recommended for items that are answered with ordinal scales, which are so popular in psychology. Since this index does not guarantee the fulfillment of assumptions such as the tau-equivalence of the items [67], its use as the only measure of the reliability of the scales is not recommended [68]. For this reason, in the present study, two additional measures of internal consistency were reported, both the omega coefficient corrected for correlated errors (0.88) and the omega coefficient for ordinal items (0.87), which show excellent levels of reliability. This evidence guarantees a lower measurement error and greater precision of the scores obtained, which is especially important in tests for clinical use [69]. These results constitute the first empirical evidence of the scale’s internal consistency, using more robust indicators of reliability.

From the perspective of the item response theory (IRT), all the items showed adequate discrimination indices; that is, they adequately differentiate between the responses of people with high behavioral activation and those with moderate or low levels. Item 4 (“It is easy for me to find reasons to enjoy life”) best takes advantage of this characteristic. This item seems especially suitable for detecting the availability of reinforcement and the person’s ability to do things that allow him to obtain positive reinforcement from his environment. This result could be helpful in intervention processes since the content of this item is related to the emotional state and, therefore, would support the effectiveness of behavioral interventions aimed at increasing reinforcement to reduce emotional stress levels [70,71]. Regarding the location indices, all the items show increasing monotonous values; that is, people with low levels of behavioral activation tend to choose the first or second category, and those with a higher trait level choose the higher categories. This pattern reflects that the content proposed in each item allows for taking advantage of all the response alternatives, and there is no loss of information.

Regarding the validity based on the relationship with other variables, in the present study, it was shown that the perception of availability of positive reinforcement from the environment (EROS-R) is negatively related to the level of anxiety (*ρ* = −0.34; *p* < 0.01) and depression (*ρ* = −0.54; *p* < 0.01). This relationship is perfectly consistent with the AC model. As previously indicated, depressive symptoms are strongly associated with less availability and less sensitivity to perceive positive environmental reinforcement [72,73]. The absence of reinforcers (or the presence of punishment) leads people to engage in avoidance behaviors that end up aggravating and making problems chronic. Negative thoughts and emotions are experienced as a barrier to engaging in antidepressant behaviors that would allow positive reinforcement from the environment [74,75]. Similarly, anxiety-associated escape and avoidance behaviors would reduce the person’s exposure to environmental reward sources [76,77].

We believe that the present study shows that the EROS-R can be successfully applied in clinical settings. The EROS-R, a brief scale containing only positive items, is likely to be particularly useful for use in patients with severe mental illness, although in this study participants with neurological deficits or other medical conditions that made it difficult to fill out the instruments were excluded. Due to similar characteristics of the disorders, such as being very focused on symptoms and fighting them, psychiatric patients could experience fatigue and difficulty concentrating on tasks or focusing attention. This could affect their ability to answer positive and negative questions alternately. Some studies show that answering scales with negative items induces more fatigue in the participants [78,79]. Additionally, people with cognitive difficulties, lower educational levels, and lower reading comprehension capacity show more biases in their response when negative items are introduced [80,81]. In this sense, the study participants did not have significant difficulties responding to this scale version. In addition, the consistent results found in the criterion validity or relation to other variables indicate that the response pattern of the participants follows the same direction (negative relationship with anxiety and depression) as the general population samples in previous studies [19,31]. This leads us to think that the EROS-R could be used to evaluate the effectiveness of treatments based on BA in clinical samples and to analyze change processes in line with new therapeutic approaches that advocate cognitive-based, therapy process-based behavior [82,83].

Despite its strengths, the study is not without limitations. First, since a non-probabilistic convenience sampling was used where males and patients with schizophrenia were predominant, this could limit the generalization of the results. On the one hand, it has been suggested that cultural factors could affect how people perceive and value the degree of reward obtained by the environment [84,85]. In this sense, it could be that men and women, who have traditionally been socialized differently, could therefore respond differently to the scale. On the other hand, people diagnosed with psychotic disorders often have a diminished or disorganized sense of self and, as a result, may have difficulty making sense of how they act and direct their own lives [59]. This aspect could introduce a differentiating factor for people suffering from other disorders. Therefore, future studies should select more representative samples of the clinical population and conduct factor invariance analyses based on the gender or psychiatric diagnosis of the participants. Second, the study used only self-report measures, and social desirability could have affected the patients’ responses. Future studies could explore using behavioral logs or peer reports to assess participants’ activity levels as a supplement to self-report measures. Third, in the present study, the temporal stability of the scale was not evaluated; therefore, it is suggested that future studies evaluate reliability through the test–retest method. Despite these limitations, we believe that the study’s results support the use of the EROS-R scale in clinical and research contexts.

## 6. Conclusions

In conclusion, the study aimed to fill a gap in measuring and investigating behavioral activation in a clinical context. The results showed that the EROS-R scale is an instrument with solid psychometric evidence based on techniques from the classical test theory and the item response theory. In addition, the study constitutes a significant contribution to the measurement in the clinical context of the processes involved in BA therapy, which can facilitate a rapid and efficient evaluation of the degree of environmental reward perceived by the patient. In this sense, we believe that the EROS-R will allow systematic monitoring of advances in therapy (including change processes), and thus, will help in evaluating the effectiveness of therapy in patients.

## Figures and Tables

**Figure 1 ijerph-19-10062-f001:**
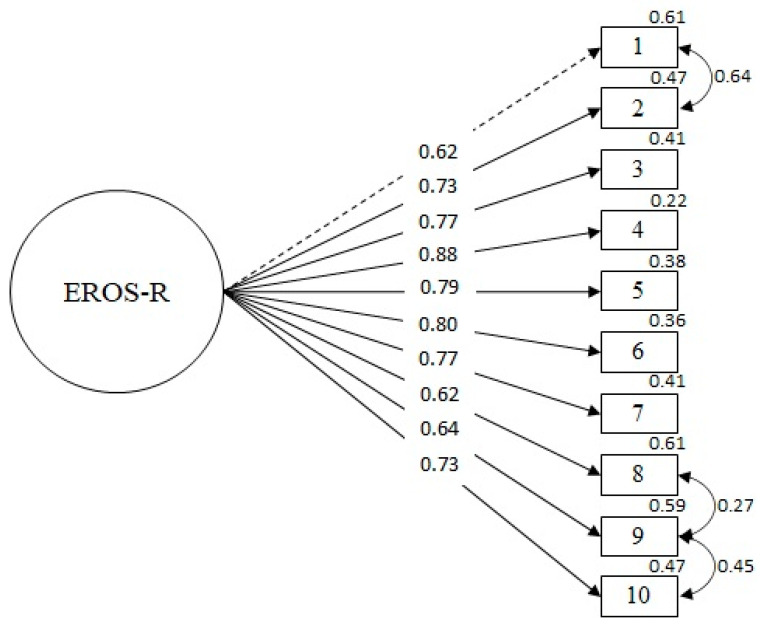
Confirmatory factor analysis of the scale. Note. All factorial weights are statistically significant (*p* < 0.01).

**Figure 2 ijerph-19-10062-f002:**
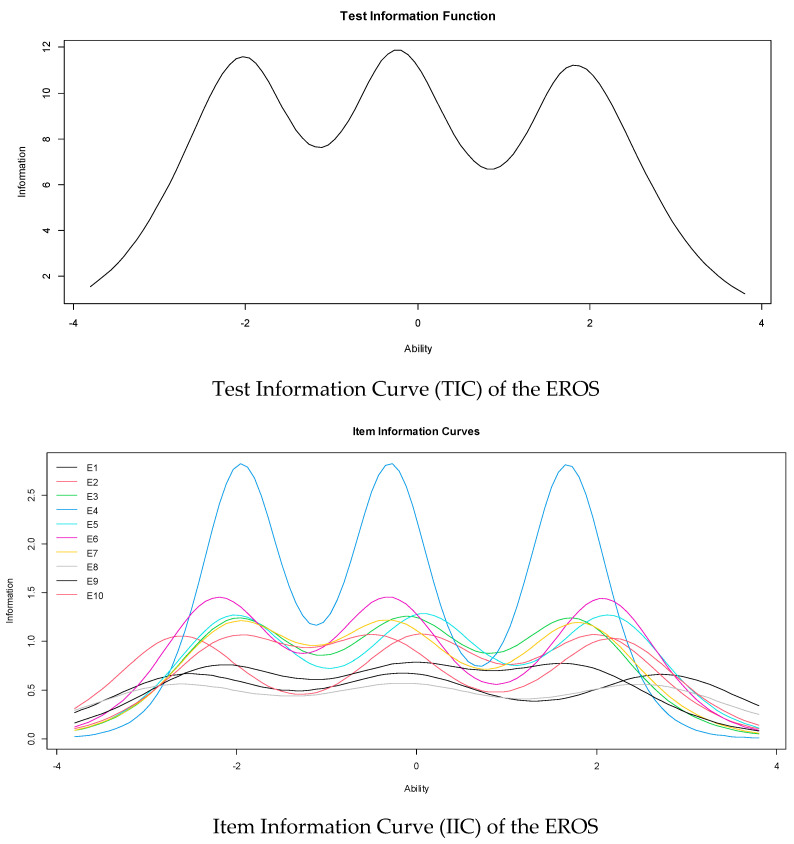
Item and test information curves.

**Figure 3 ijerph-19-10062-f003:**
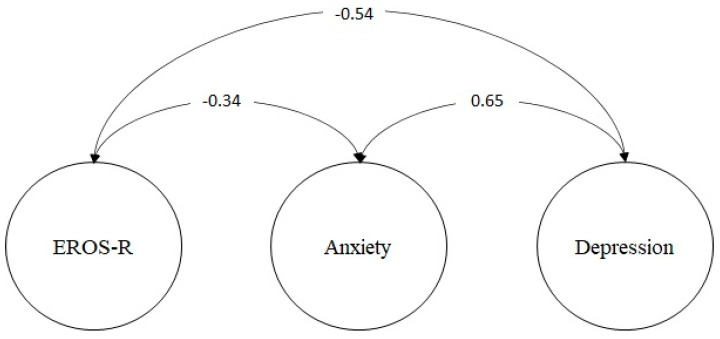
Relationship model with other constructs.

**Table 1 ijerph-19-10062-t001:** Polychoric correlation matrix and descriptive analysis of the items.

Items	E1	E2	E3	E4	E5	E6	E7	E8	E9	E10
E1	1									
E2	0.80	1								
E3	0.50	0.62	1							
E4	0.63	0.65	0.67	1						
E5	0.55	0.60	0.55	0.73	1					
E6	0.41	0.53	0.52	0.73	0.70	1				
E7	0.43	0.52	0.58	0.64	0.59	0.62	1			
E8	0.30	0.42	0.56	0.48	0.41	0.51	0.51	1		
E9	0.40	0.35	0.53	0.56	0.42	0.56	0.53	0.56	1	
E10	0.43	0.52	0.53	0.58	0.50	0.57	0.66	0.53	0.70	1
*M*	2.60	2.59	2.63	2.68	2.52	2.63	2.66	2.56	2.63	2.70
*SD*	0.64	0.68	0.70	0.63	0.67	0.62	0.67	0.63	0.73	0.59
g1	−0.06	0.22	−0.06	−0.13	−0.23	−0.26	−0.46	0.03	0.19	−0.83
g2	−0.19	−0.31	−0.19	−0.07	−0.16	−0.03	0.24	−0.25	−0.43	0.84

Note. *M* = mean; *SD* = standard deviation; g1 = skewness; g2 = kurtosis.

**Table 2 ijerph-19-10062-t002:** Discrimination parameters and location for the items of each dimension.

Items	Item Parameters				Item Fit			Model Fit Indices					
	a	b_1_	b_2_	b_3_	S-X2 (df)	*p*	RMSEA.S-X2	M2 (df)	*p*	RMSEA	SRMSR	TLI	CFI
E1	1.62	−2.55	−0.12	2.75	13.65 (13)	0.399	0.015	29.05 (15)	0.016	0.064	0.068	0.95	0.97
E2	2.05	−2.62	0.04	2.08	14.60 (14)	0.406	0.014						
E3	2.21	−1.99	−0.11	1.74	24.32 (16)	0.083	0.048						
E4	3.35	−1.95	−0.29	1.67	13.96 (12)	0.303	0.027						
E5	2.25	−2.02	0.08	2.14	20.55 (13)	0.082	0.051						
E6	2.40	−2.21	−0.31	2.07	21.09 (13)	0.071	0.052						
E7	2.17	−1.99	−0.29	1.81	28.13 (14)	0.014	0.060						
E8	1.48	−2.68	−0.14	2.52	15.74 (16)	0.472	0.000						
E9	1.72	−2.19	−0.08	1.74	17.09 (18)	0.516	0.000						
E10	2.02	−2.03	−0.41	2.18	21.94 (14)	0.080	0.050						

*Note.* a = discrimination parameters; b = location parameters.

**Table 3 ijerph-19-10062-t003:** Relationship of EROS-R with other variables.

Structural Model							
χ^2^	df	*p*	RMSEA	CI 90%	SRMR	CFI	TLI
581.02	293	0.000	0.066	0.058–0.074	0.086	0.92	0.91
Measurement Models							
Items	EROS-R		Anxiety		Depression		
	λ (error)		λ (error)		λ (error)		
1	0.65 (0.57)		0.73 (0.47)		0.56 (0.68)		
2	0.70 (0.51)		0.75 (0.43)		0.72 (0.49)		
3	0.72 (0.47)		0.77 (0.40)		0.67 (0.54)		
4	0.89 (0.22)		0.77 (0.41)		0.65 (0.57)		
5	0.79 (0.37)		0.78 (0.39)		0.58 (0.66)		
6	0.80 (0.36)		0.58 (0.66)		0.59 (0.66)		
7	0.74 (0.44)		0.69 (0.51)		0.52 (0.72)		
8	0.61 (0.63)				0.51 (0.73)		
9	0.70 (0.51)				0.81 (0.34)		
10	0.79 (0.38)						

*Note:* χ^2^ = chi-square test; gf = degrees of freedom; *p* = *p*-value; RMSEA = root mean square error of approximation; SRMR = standardized root mean squared residual; CFI = comparative fit index; TLI = Tucker–Lewis index; λ = factor loading.

## Data Availability

Upon request, authors are prepared to send relevant documentation or data in order to verify the validity of the results.

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
