# Peer review of "Functioning of the EROS-R Scale in a Clinical Sample of Psychiatric Patients: New Psychometric Evidence from the Classical Test Theory and the Item Response Theory"

_ijerph, 2022, doi:10.3390/ijerph191610062_

Round 1

Reviewer 1 Report

The article is of interest because it provides new information on the psychometric properties of the EROS-R scale in Spanish when evaluating a clinical population. To assess the clinical properties, the authors use techniques from both classical test theory and item response theory, which provide relevant information regarding their performance and endow the analyzes with a certain methodological rigor.

However, some gaps have been found that need to be clarified:

1. Although in general terms the introduction summarizes the relevant information, I think it is necessary to specifically mention some findings of comparisons between groups, for example, differences between men and women, depressed and non-depressed people, or age groups, among other variables. In addition, it would be very convenient to explain in greater detail why it is important to assess the role of positive reinforcement specifically in the clinical setting and for mental health therapy. 

2. In the introduction, psychometric information regarding the instrument and explanatory information on item response theory is not necessary. If the authors want to keep these paragraphs they should be in the methods section.

3. Regarding the participants, although the sample size can be considered adequate for the applications of classical test theory, no information is presented on the adjustment of the graduated response model to the data or the adjustment of the items, and therefore, this may raise doubts about the adequacy of the sample size for the application of IRT-based analyzes under the said model.

4. In the description of the participants, information is missing, such as the data collection period, the type of mental health care that the participants received (outpatient or inpatient), the time elapsed since the diagnosis (this variable is of great relevance because it can affect the results: patients with less time since diagnosis may be more reactive and more unstable), region of residence (urban or rural), or type of therapy received by the participants. Information on the number of participants for each diagnosis is also missing.

5. In relation to the instruments, the psychometric properties of the other scales used (GAD-7 and PHQ-9) or their Spanish version are missing.

6. In relation to the procedure, it is not sufficiently clear if the scale was self-administered by the participants or if the psychologist was reading each of the questions. If the items were read by the psychologists, was there a training period for them? What did it consist of? There is no information regarding the application session of the scale (was it carried out as part of a consultation already established for the participant? or was it a specific session for the administration of the instrument?). Also missing is the mention of how the list of possible participants was established. In the manuscript it is mentioned that the sampling was not random, but what specific type of sampling was followed?

7. For the data analysis, there is a lack of information on the adequacy of the IRT model applied for the analysis, in addition to the fulfillment of assumptions; how was it guaranteed that the data adjusted to the model used?

8. Regarding the results, figure 2 does not correspond to the characteristic curves of the items and the test, but to the information function. This needs to be specified because it causes confusion. Regarding the results of this figure, it is necessary to mention the implications that the fact that item 4 provides more information than the other items and that the maximization of this information is presented in three intervals of the skill level may have for the measure. If the authors want to annex the ICC figures, these could be included as supplementary material.

9. A discussion is needed on the implications of the correlations obtained between the errors of the items and their possible effect on the precision of said items. Is there any relationship between these items and those that were converted from reverse to positive wording?

10. It is very convenient to include an analysis of the differential functioning of the item or of the invariance of the scale since it is a relevant step if the testing standards are followed. Since the information provided regarding validity is related to other variables and internal structure, the analysis would be enriched if a DIF analysis is applied, probably between diagnoses, gender, or other variables in which differences between groups have been reported.

11. A brief mention of the limitations and strengths of the study carried out is missing, a way to recognize some shortcomings of the study may be of great interest to researchers and to those users interested in promoting the use of this scale.

12. Finally, it is not clear how, based on the results obtained regarding the evidence of relationship with other variables, it can be concluded that the participants responded in a similar way to the general population samples used in other studies. To do this, it is necessary to delve into both response patterns and collect information on the estimation of item parameters. Although the directions of the expected relationships are confirmed, this alone does not inform about the similarity of responses between groups.

Author Response

Thank you very much for the observations; they helped a lot for a better article presentation. All changes made are marked in red. Below I attach the document with the answers to each observation.

Reviewer 2 Report

This is an interesting manuscript reporting on the psychometric properties of the Environmental Reward Observation Scale (EROS-R)- a short scale containing only positive items- in psychiatric patients. The use of reliable and valid assessment instruments facilitates clinical practice and research work. The article argues convincingly for the need to test the psychometric evidence of the EROS-R scale in clinical samples. 

 Below I have listed aspects to improve:

- It is recommended to develop the abbreviation TCT and describe it at some point what the Classical Test Theory is. I think it can be interesting since it appears in the title of the study.

In the same way, it is recommended to develop at some point the name of the indices RMSEA, SRMR, CFI and TLI.

-   I miss that the authors provided the EROS-R version in an appendix.

Methods

Participants: It is recommended to clarify if the participants are inpatients/outpatients. One of the objectives of the study is to estimate the reliability of the scale in hospitalized psychiatric patients. This aspect is not clear.

It is recommended to further specify the characteristics of the subjects in the sample (symptoms of depression, cognitive or reading comprehension difficulties...)

Results

Could the Reliability results be illustrated with a graph?

Figure 2. Item Information Curves title does not display very well.

Table 3. Check relationship between the EROS scale and the level of anxiety in item 8, 9,10. Is there no value?.

Add in the legend of Table 3 the abbreviations and acronyms developed (x2, gl, p, RMSEA...)

References

-Reference 15 shows the abbreviated title of the journal. The complete name is Health and quality of life outcomes.

-If the referenced books have a DOI link, it should be added.

Author Response

(The authors gave the same response as above.)

Reviewer 3 Report

Title

Adequate

Summary

Adequate

Introduction

Line 39: it is advisable to include some more current quotes from the behavioural line.

Materials and methods

It is advisable to indicate initially the type of study and its design.

Line 119: it is advisable to indicate, in addition to the % of men and women, the exact number of each of them.

Results

Line 215 and 251: Figures 1 and 2 should be named above the figure, not below it.

Line 258: Table 3 should not be presented on 2 different pages.

Line 265: Figure 3 should be named above the figure, not below it.

Discussion and conclusion

The Discussion section should start by presenting the general objective of the study.

References

Line 392: The pagination should not be presented inside brackets.

Line 392: The name of the journal is written in italics.

Line 396: The name of the journal is written in italics.

A check should be made to ensure compliance with the APA 7th edition, as the errors mentioned above are repeated in some references.

Some references do not include the link to locate them.

Author Response

(The authors gave the same response as above.)
